# Short-Chain Fatty Acids Weaken Ox-LDL-Induced Cell Inflammatory Injury by Inhibiting the NLRP3/Caspase-1 Pathway and Affecting Cellular Metabolism in THP-1 Cells

**DOI:** 10.3390/molecules27248801

**Published:** 2022-12-12

**Authors:** Chengxue Yi, Wen Sun, Longkun Ding, Man Yan, Chang Sun, Chenguang Qiu, Dongxu Wang, Liang Wu

**Affiliations:** 1School of Medical Technology, Zhenjiang College, Zhenjiang 212028, China; 2Department of Critical Care Medicine, Jurong People’s Hospital, Jurong 212499, China; 3Department of Laboratory Medicine, School of Medicine, Jiangsu University, Zhenjiang 212013, China; 4Stomatology Department, Zhenjiang First People’s Hospital, Zhenjiang 212003, China; 5School of Grain Science and Technology, Jiangsu University of Science and Technology, Zhenjiang 212100, China

**Keywords:** short chain fatty acids, oxidized low-density lipoprotein, metabolomics, anti-inflammation, cellular metabolism

## Abstract

Short-chain fatty acids (SCFAs) are important anti-inflammatory metabolites of intestinal flora. Oxidized low-density lipoprotein (ox-LDL)-induced macrophage activation is critical for the formation of atherosclerosis plaque. However, the association between SCFAs and ox-LDL-induced macrophage activation with respect to the formation of atherosclerosis plaque has not yet been elucidated. The present study investigated whether SCFAs (sodium acetate, sodium propionate, and sodium butyrate) can affect ox-LDL-induced macrophage activation and potential signaling pathways via regulation of the expression of the NLRP3/Caspase-1 pathway. Using human monocyte-macrophage (THP-1) cells as a model system, it was observed that ox-LDL not only induced cell inflammatory injury but also activated the NLRP3/Caspase-1 pathway. The exogenous supplementation of three SCFAs could significantly inhibit cell inflammatory injury induced by ox-LDL. Moreover, three SCFAs decreased the expression of IL-1β and TNF-α via the inactivation of the NLRP3/Caspase-1 pathway induced by ox-LDL. Furthermore, three SCFAs affected cellular metabolism in ox-LDL-induced macrophages, as detected by untargeted metabolomics analysis. The results of the present study indicated that three SCFAs inhibited ox-LDL-induced cell inflammatory injury by blocking the NLRP3/Caspase-1 pathway, thereby improving cellular metabolism. These findings may provide novel insights into the role of SCFA intervention in the progression of atherosclerotic plaque formation.

## 1. Introduction

Cardiovascular disease caused by atherosclerosis is a kind of chronic inflammatory disease that is the leading cause of mortality worldwide. An atherosclerotic process begins with the activation of the endothelium, followed by the accumulation of lipids, which causes vessels to narrow and inflammatory pathways to be activated [1]. Therefore, hyperlipidemia is an important characteristic of patients with atherosclerosis and chronic inflammation. The low-density lipoprotein (LDL) is a lipoprotein particle that can carry cholesterol into peripheral tissue cells and be oxidized to oxidized low-density lipoprotein (ox-LDL) [2]. Macrophages engulfing ox-LDL accumulate in the arterial wall and transform into foam cells, releasing large amounts of pro-inflammatory cytokines and chemokines to induce inflammatory damage in the vessel wall, accelerating the formation of atherosclerosis plaque [3,4].

It is widely known that increasing the diet of *Lactobacillus* and dietary fiber can effectively prevent the formation of atherosclerosis plaque [5,6,7,8]. For instance, high fat diet-induced enhanced atherosclerotic plaque amount was decreased by both whole bean and isolated fiber fraction, which is associated with the higher formation of cecal short-chain fatty acids (SCFAs) in ApoE^−/−^ mice [8]. In the gut, SCFAs produced by fermenting nondigestible carbohydrates are absorbed by the microbiota [9]. It has been found that SCFAs may play an important role in glucose homeostasis, lipid metabolism, and immune function through free fatty acid receptors 2 and 3 (FFAR 2/3). As a result, pharmacological interventions targeting this receptor-mediated pathway have been widely studied for cardiovascular disease [9]. There is recent evidence suggesting that the NLR family pyrin domain containing 3 (NLRP3) inflammasome is responsible for maturing and secreting precursor interleukin 18 (IL-18)/IL-1β by activating caspase-1 [10]. NLRP3-driven inflammatory responses are implicated in atherosclerosis plaque formation, according to emerging evidence [11,12]. For instance, selective NLRP3 inhibitor MCC950 can hinder atherosclerosis development by attenuating inflammatory responses and macrophage activation in ApoE^−/−^ mice and ox-LDL-induced THP-1-derived macrophages [13]. Therefore, the inhibition by ox-LDL-induced NLRP3 activation and inflammatory response in macrophages are key to intervening in the formation of atherosclerosis plaque. Recently, a group of SCFAs with significant anti-inflammatory activity that is closely related to the activation of the NLRP3 inflammasome and carbon atoms less than 6, including acetic acid (NaAc), propionic acid (NaPc), and butyric acid (NaB), were produced mainly from dietary fiber by fermentation of certain anaerobic bacteria in the colon [14,15,16,17].

Increasing the production of SCFAs in the intestine can effectively reduce the incidence of cardiovascular diseases induced by hyperlipidemia [18,19], but the association between SCFAs and ox-LDL-induced macrophage activation with respect to the formation of atherosclerosis plaque remains poorly understood. In this study, we investigated the inhibitory effects of three SCFAs (NaAc, NaPc, and NaB) on ox-LDL-induced inflammation in human THP-1 derived macrophages (THP-1 cells) and further studied the anti-inflammatory mechanisms by untargeted metabolomics techniques. Here, in ox-LDL-stimulated THP-1 cells, our data are the first to confirm that three SCFAs suppress inflammation events via inhibiting NLRP3 activation. Then, untargeted metabolomics analysis verified that three SCFAs affected cellular metabolism in ox-LDL-induced macrophages.

## 2. Results

### 2.1. SCFAs Inhibited the Expression of Pro-Inflammation Cytokines

Firstly, the expression pattern of inflammation cytokines in ox-LDL-stimulated THP-1 cells was determined by qPCR assay. Compared with the NC group, ox-LDL could up-regulate the mRNA expressions of pro-inflammation cytokines IL-1β and TNF-α while down-regulating the mRNA expressions of anti-inflammation cytokines IL-10 and TGF-β in ox-LDL-stimulated THP-1 cells (*p* < 0.05). Compared with the ox-LDL group, SCFAs, including NaAc, NaPc, and NaB, significantly inhibited the mRNA expressions of IL-1β and TNF-α and restored the mRNA expressions of IL-10 and TGF-β (*p* < 0.05) in ox-LDL-stimulated THP-1 cells (Figure 1).

### 2.2. SCFAs Inhibited NF-κB p65 Protein Nuclear Translocation

Canonical NLRP3 inflammasome activation is usually a two-step process: priming and activation. Prior to activation, an inflammatory ‘priming’ stimulus is required. It is thought that NLRP3 and pro-IL-1β induction are both required for nuclear factor kappa B (NF-κB) priming at the downstream end of pattern recognition receptors [20]. As part of our investigation of SCFAs’ effect on NLRP3 in THP-1 cells in response to ox-LDL stimulation, we measured protein amounts of NF-κB p65. Compared with the NC group, NF-κB p65 protein expression was significantly increased in the nucleus and decreased in the cytoplasm in the ox-LDL group (*p* < 0.05). Compared with the ox-LDL group, NF-κB p65 protein expression was significantly reduced in the cell’s nucleus. In contrast, the expression in cell cytoplasm was increased dramatically in the NaAc, NaPc, and NaB treatment groups (*p* < 0.05) (Figure 2).

### 2.3. SCFAs Inhibited the Production of ROS in THP-1 Cells

Recent evidence suggests that ROS activates NLRP3 by changing the redox state, which makes it possible to detect changes in the level of ROS in cells. Our result showed that ROS production was significantly increased in the ox-LDL group compared to the NC group, while it was decreased in the NaAc, NaPc, and NaB treatment groups compared to the ox-LDL group in ox-LDL-stimulated THP-1 cells (*p* < 0.05) (Figure 3).

### 2.4. SCFAs Inhibited NLRP3 Inflammasome Activation in THP-1 Cells

The activation of NF-κB p65 and the enhancement of ROS are the important molecular basis for the activation of NLRP3 inflammasome. Next, we investigated whether inhibition of NF-κB p65 activation would inhibit NLRP3 inflammasome activation caused by ox-LDL in THP-1 cells. Our result showed that compared with the NC group, the expressions of NLRP3 and Caspase-1 in THP-1 cells were significantly increased in the ox-LDL group (*p* < 0.05). Moreover, compared with the ox-LDL group, the expressions of NLRP3 and Caspase-1 were significantly decreased in the NaAc, NaPc, and NaB treatment groups (*p* < 0.05) (Figure 4).

### 2.5. PCA and OPLS-DA of Cellular Metabolites

The UPLC-IMS Q-Tof MS/MS data of cell samples from the NC, ox-LDL, NaAc, NaPc, and NaB groups were subjected to PCA (*n* = 3). The sample data of the NC and ox-LDL groups, ox-LDL and NaAc groups, NaPc, and NaB groups were significantly distinguished (Figure 5A). The three fit indices of the OPLS-DA model were *R*^2^*X* = 0.854, *R*^2^*Y* = 0.982, and *Q*^2^** = 0.808, which showed that the OPLS-DA model could be used for subsequent analysis. The NC and ox-LDL groups, ox-LDL and NaAc groups, ox-LDL and NaPc groups, and ox-LDL and NaB groups could be easily distinguished from each other. However, the NaAc, NaPc, and NaB groups could not be easily distinguished from each other, indicating that the metabolites of the NC and ox-LDL groups, ox-LDL and NaAc groups, ox-LDL and NaPc groups, and ox-LDL and NaB groups were significantly different. Although, the metabolites in the NaAc, NaPc, and NaB groups were not significantly different (Figure 5B).

### 2.6. Heat Map and Volcano Plot Analysis of Differential Metabolites

The subgroup clustering heat map shows the trend of the differential metabolite content in each group. We observed significant changes in differential metabolite profiles between the NC and ox-LDL groups. We also found significant changes in differential metabolite profiles between the ox-LDL group and other SCFAs groups, including the NaAc, NaPc, and NaB groups. However, the changes in differential metabolite profiles among the NaAc, NaPc, and NaB groups were insignificant (Figure 5C). The results of volcano plot analysis showed that the differential metabolites were significantly altered between the NC and ox-LDL groups, ox-LDL and NaAc groups, ox-LDL and NaPc groups, and ox-LDL and NaB groups, using VIP > 1 and *p* < 0.05 as the boundary (Figure 6A). Screening the potential of different metabolites based on VIP > 1 and *p* < 0.05 and the top 15 different metabolites related to ox-LDL, NaAc, NaPc, and NaB treatment are shown in Table 1, Table 2 and Table 3, respectively.

### 2.7. KEGG Enrichment Analysis of Metabolite Pathways

The KEGG online database was used to enrich the associated metabolic pathways to explore further potential mechanisms (Figure 6B). The three SCFAs could affect the sphingolipid metabolism pathway of THP-1 cells. NaAc and NaPc could affect pentose and glucuronate interconversions of THP-1 cells, including pentose and glucuronate interconversions, sphingolipid metabolism, phenylalanine, tyrosine, tryptophan biosynthesis, and phenylalanine metabolism.

## 3. Discussion

Chronic uncontrollability inflammation in the vascular wall is an initial factor for atherosclerosis [21]. Excessive ox-LDL in the body can trigger an inflammatory response, activate macrophages, and up-regulate the expressions of pro-inflammation cytokines (such as IL-1β and TNF-α), which ultimately causes endothelial cell damage and atherosclerosis plaque formation [22]. A number of inflammatory mediators are secreted by macrophages in the subcutaneous space when ox-LDL is induced, including interferons and interleukins, which contribute to the development and initiation of atherosclerosis [23,24]. A previous study indicated that Ox-LDL activates the NF-κB pathway and leads to the inflammation of monocytes [25]. Here, we also verified that 50 mg/mL of ox-LDL contributed to an inflammatory response in THP-1 cells by up-regulating the expression of ROS, IL-1β, and TNF-α of THP-1 cells by activation of the NLPR3 inflammasome.

Multiple protein complexes are known as inflammasomes. NLRP3 is one of them and regulates inflammatory responses [26]. The inflammasomes trigger pro-inflammatory signals and Caspase-1-dependent pyroptosis, leading to the release of mature IL-1β and IL-18 via NF-κB priming [26]. Inflammatory responses are triggered by the activation of NLRP3 inflammasomes, which convert deactivated Caspase-1 precursor to active Caspase-1, which leads to the maturation, activation, and signaling of Caspase-1 [27]. Some studies have suggested that the average concentrations of SCFAs in mouse serum are approximately 200 μmol/L [28]. As a result of our preliminary tests, we found no significant effect of the three SCFAs at a concentration of 200 mol/L on the proliferation of THP-1 cells [29]. In this study, we found that the three SCFAs could effectively inhibit the ox-LDL-induced inflammation of THP-1 cells. ROS production and the expressions of NLRP3, Caspase-1, TNF-α, and IL-1β were significantly decreased when treated with the three SCFAs compared to the ox-LDL group. In addition, the three SCFAs could significantly suppress the activation of the NF-κB signaling pathway. The above results indicate that the three SCFAs inhibit ox-LDL-induced inflammation via the ROS/NF-κB/NLRP3 pathway. Activating the NF-κB signaling pathway and NLRP3 inflammasome is a key step in ox-LDL-inducing macrophage activation [30]. In the hyperlipidemic rat model, abnormal activation of the NF-κB pathway and a significant reduction in the secretion of proinflammatory cytokines were observed in rats injected with an inhibitor of IkappaB kinase (IKK) [31]. The NLRP3 inflammasome was also activated in the obese population with hyperlipidemia. The expressions of NLRP3 and IL-1β were significantly increased in obese models with hyperlipidemia compared to those with normal lipids [32].

The results of the metabolomic study show that prostaglandin E2 (PGE2) and reduced glutathione (GSH) are significantly up-regulated in THP-1 cells after pretreatment with NaPc. PGE2 is synthesized from arachidonic acid in human cells via the cyclooxygenase pathway. It has been found that low concentrations of PGE2 (0.01–1 μmol/L) can promote platelet aggregation and atherogenesis. Compared to low levels of PGE2, high levels (>10 μmol/L) can inhibit platelet ADP release and collagen synthesis and reduce platelet aggregation and atherosclerosis [33]. Similarly, in lipopolysaccharide-stimulated human neutrophils and monocytes, SCFAs (propionate and butyrate) at 0.2–20 mmol/L can increase the production of PGE2 [34]. We hypothesized that NaAc and NaPc might protect blood vessels and prevent atherosclerosis through the above mechanism.

As an antioxidant, reduced GSH plays an important role in maintaining intracellular redox balance and eliminating ROS. Some studies have found that dihydromyricetin, xanthoangelol, or isothiocyanates protect ox-LDL-induced human umbilical vein endothelial cells from inflammatory injury and endothelial dysfunction by increasing GSH content [35,36,37]. These results suggest that enhanced GSH synthesis by SCFAs plays an important role in reducing the inflammatory injury of macrophage cells, which may be related to the inhibition of NLRP3 activation via regulating redox state. Further research is needed to determine whether SCFAs can activate the nuclear factor erythroid 2-related factor 2 (Nrf2), which is a nuclear factor responsible for regulating GSH synthesis, particularly during cell stress and inflammatory injury. There is some evidence to suggest that SCFAs-mediated activation of the Nrf2 defense pathway protects against oxidative stress in vitro [38,39,40,41]. For now, how SCFAs activate the Nrf2 pathway in endothelial cells remains unclear, though it may be related to the FFAR3. In neuron cells, SCFAs were found to activate the Nrf2 pathway via specificity protein 1 (SP1)-mediated p21 expression through a sodium-coupled monocarboxylate transporter 1 [42]. In addition, the Nrf2-targeted thioredoxin-1 protein inactivates thioredoxin-interacting proteins (Txnip), which may promote ROS accumulation and activate the NLRP3 inflammasome [43]. Therefore, the inhibition of Txnip by Nrf2-targeted thioredoxin-1 is involved in NLRP3 inflammasome inactivation in a redox-dependent manner. According to our results, SCFAs are likely to inactivate NLRP3 by up-regulating Nrf2-mediated Trx1, which directly binds to Txnip. This hypothesis needs further research.

Ceramide is one of the main components of bilayer lipid rafts found in cell and organelle membranes consisting of sphingolipids, with sphingomyelin as their backbones [44]. Sphingolipids such as ceramide can maintain the structural stability of cell and organelle membranes and play important roles in physiological activities such as cell proliferation, differentiation, and apoptosis processes [45]. Pro-inflammatory factors such as bacterial lipopolysaccharide and TNF-α can activate the synthesis of ceramide and enrich the metabolic pathway of sphingolipids. The activation of sphingolipid metabolic pathways positively correlates with atherogenesis, which might be due to high inflammatory states activating the sphingolipid metabolism in hyperlipidemic mice [46]. Recently, Chen et al. found that Huanglian Jiedu decoction can alleviate acute lung injury by inhibiting the NLRP3/caspase-1 pathway via the sphingolipid pathway in rats [47]. In our study, we also found that the sphingolipids metabolic pathway of THP-1 cells was activated by ox-LDL in vitro and inhibited by NaPc. We hypothesized that the SCFAs could inhibit macrophage inflammation through sphingolipid’s metabolic pathway. This should have been further explored, which is a shortcoming of our study.

## 4. Materials and Methods

### 4.1. Cell Line and Experimental Design

THP-1 cells (National Collection of Authenticated Cell Cultures, Shanghai, China) were cultured in an RPMI 1640 medium (Biological Industries, Kibbutz, Israel) containing 10% fetal bovine serum (Biological Industries, Kibbutz, Israel) at 37 °C and 5% CO_2_, and logarithmic growth phase cells were taken for the experiment. The cells were inoculated in 6-well cell culture plates at a density of 5 × 10^5^/well. In the NC group, the cells were not treated. In the ox-LDL group, the cells were exposed to ox-LDL (Yiyuan Biotechnology Co., Ltd., Guangzhou, China) at a final concentration of 50 mg/L, which caused inflammation for 24 h. The cells in the three SCFAs treatment groups were pre-treated with NaAc, NaPc, and NaB (Bomei Biotechnology Co., Ltd., Hefei, China) at a final concentration of 200 μmol/L, respectively, for 24 h, and then stimulated with 50 mg/L ox-LDL for 24 h. The cells were collected at the end of the experiment for subsequent experiments.

### 4.2. Quantitative PCR (qPCR) Assay

The total RNA of THP-1 cells was extracted by Trizol reagent (Vazyme biotechnology, Nanjing, China) and reverse transcribed to cDNA according to the RNA assay kit (Vazyme biotechnology, China) instructions for subsequent detection by SYBR qRT-PCR Master Mix (Vazyme biotechnology, Nanjing, China) using a real-time quantitative PCR instrument (Bio-Rad, Hercules, CA, USA). The total reaction system was 20 μL and the reaction was pre-denatured at 95 °C for 1 min, denatured at 95 °C for 5 s, annealed at 58 °C for 20 s, and extended at 72 °C for 30 s. The reactions were performed for a total of 40 cycles. The sequences of IL-1β (F: 5′-CCTGTCCTGCGTGTTGAAAGA-3′, R: 5′-GGGAACTGGGCAGACTCAAA-3′), TNF-α (F: 5′-AATGGCGTGGAGCTGAGA-3′, R: 5′-TGGCAGAGAGGAGGTTGACC-3′), IL-10 (F: 5′-TCTCCGAGATGCCTTCAGCAGA-3′, R: 5′-TCAGACAAGGCTTGGCAACCCA-3′), TGF-β (F: 5′-GCGGACTACTATGCTAAAGAGG-3′, R: 5′-GTAGAGTTCCACATGTTGCTCC-3′), and β-actin (F: 5′-CATTGCTGACAGGATGCAGAAGG-3′, R: 5′-TGCTGGAAGGTGGACAGTGAGG-3′) were derived from the National Center for Biotechnology Information and synthesized by Generay Biotechnology (Shanghai, China). Each experiment group was repeated three times and the relative expressions of mRNA were calculated by equation 2^−ΔΔ^Ct, using the β-actin gene as an internal reference.

### 4.3. Western Blotting Analysis

The cells were washed twice with a cooling PBS buffer and lysed in 100 μL of RIPA lysis solution (Beyotime, Shanghai, China) containing a PMSF protease inhibitor for 30 min. The supernatant was centrifuged at 16,000× *g* for 10 min, and protein concentration was detected by the BAC method. The protein samples were separated by SDS-PAGE gel electrophoresis at a constant pressure of 80 V for 120 min and then transferred to a PVDF membrane (0.45 μm, Millipore, St. Louis, MO, USA) at 200 mA for 90 min. The PVDF membranes were blocked with 5% skim milk for 1 h at room temperature and then incubated overnight at 4 °C with primary antibodies against NF-κB p65 (1:5000, Boster Biological Technology, Wuhan, China), NLRP3 (1:2000, Boster Biological Technology, Wuhan, China), Caspase-1(1:5000, Boster Biological Technology, Wuhan, China), and β-actin (1:10,000, Boster Biological Technology, Wuhan, China). The PVDF membranes were washed three times with a TBST buffer and incubated with HPR-conjugated Goat Anti-Rabbit/Mice IgG (1:10,000, Boster Biological Technology, China) at room temperature for 1 h. An enhanced chemiluminescence reagent (Merck, Millipore, Darmstadt, Germany) was used and the grayscale values were analyzed by Image J software. Each experiment was repeated three times.

### 4.4. Cellular Reactive Oxygen Species (ROS) Detection

The cells were washed with a serum-free culture medium and resuspended. The 2,7-dichloride-hydrofluorescein diacetate (DCFH-DA) probe solution (Beyotime Biotechnology, Shanghai, China) at 10 μmol/L was incubated with cells at 37 °C for 20 min. Then, the cells were washed with a serum-free culture medium to remove the uncombined DCFH-DA probe and then resuspended in a sterile PBS buffer for flow cytometry analysis (Beckman Coulter, Brea, CA, USA). Each experiment was repeated three times.

### 4.5. Cellular Metabolites Analysis

THP-1 cells were resuspended in 2 mL 80% (*v*/*v*) methanol and the mixtures were placed at −80 °C for 20 min. After being sonicated for 10 min they were incubated at −20 °C for 1 h. The supernatant was isolated at 14,000 r/min for 5 min and dried by freeze-drying. The lyophilized cell extracts were solubilized with 60 μL of 0.1% formic acid and then fully solubilized by vortex shaking for 10 min, followed by centrifugation at 16,000 × *g* for 15 min. The supernatant was sent to the Testing Center of Yangzhou University for untargeted metabolomics analysis by UPLC-QTof-MS/MS.

### 4.6. UPLC-Qtof-MS/MS Data Collection and Analyzation

Data collection and molecular formula matching to determine potential markers were conducted using Unifi software. Progenesis QI software was used for peak match, peak aligns, and control sample names and peak intensities data matrixes. Then, the data were entered into the software for multivariate statistical analysis, using unsupervised principal component analysis (PCA) and orthogonal least squares discrimination analysis (OPLS-DA) to evaluate whole clustering trends and visualize their distributions. Variable importance in the projection (VIP) reflected the contribution of the analyzed variables to the OPLS-DA model, and potential differential metabolites were screened based on the criteria of VIP > 1, *p* < 0.05. The identifications of these differential metabolites were based on mass spectrometry data, which were searched and confirmed using the Human Metabolome Database (http://www.hmdb.ca/, accessed on 10 June 2022). The relevant metabolic pathways of potential biomarkers were identified based on the KEGG online database (https://www.kegg.jp/, accessed on 10 June 2022).

### 4.7. Data Processing and Analysis

The experimental data results were expressed as means ± standard deviations (SDs) and statistically analyzed using SPSS 22.0 statistical software. The overall mean was tested using a one-way ANOVA test and the multiple comparisons of the sample means between groups were tested using the LSD-*t* method; differences were considered statistically significant at *p* < 0.05 and Graphpad Prism 9.2 was used for graphing.

## 5. Conclusions

In summary, we observed that the three SCFAs could inhibit ox-LDL-induced macrophage inflammatory responses and exert anti-inflammatory effects via the NF-κB/NLRP3 signaling pathway (Figure 7). The three SCFAs might also inhibit inflammation by affecting the macrophage sphingolipid metabolism pathway. This study is important as it suggests a possible link between SCFA administration and ox-LDL-induced macrophage activation. Our findings provide a theoretical basis for using therapies such as lactic acid bacteria products or SCFAs for atherosclerosis prevention and adjunctive treatment.

## Figures and Tables

**Figure 1 molecules-27-08801-f001:**
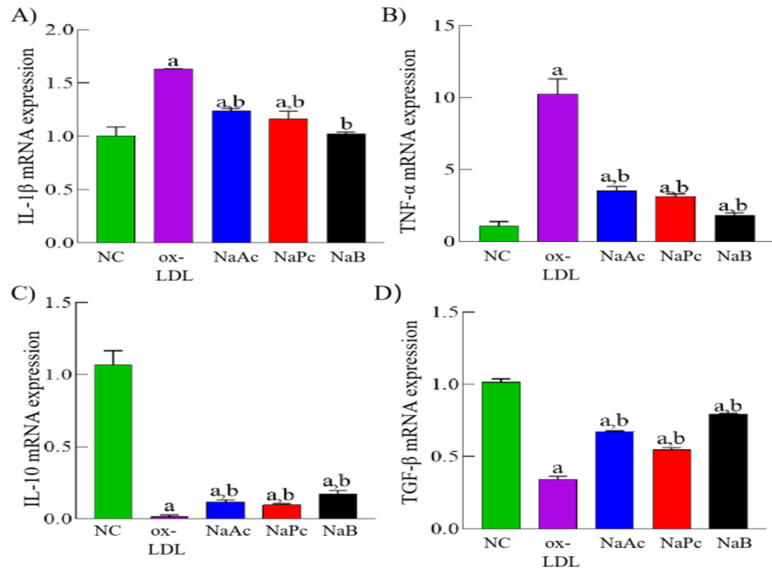
The pro-inflammation cytokines expressions of THP-1 cells. The THP-1 cells were pre-treated with 50 mg/L ox-LDL for 24 h and then treated with 200 μmol/L NaAc, NaPc, and NaB for 24 h. The cytokines expressions were determined by qPCR assay. (**A**) IL-1β. (**B**) TNF-α. (**C**) IL-10. (**D**) TGF-β. ^a^
*p* < 0.05 vs. NC group. ^b^
*p* < 0.05 vs. ox-LDL group.

**Figure 2 molecules-27-08801-f002:**
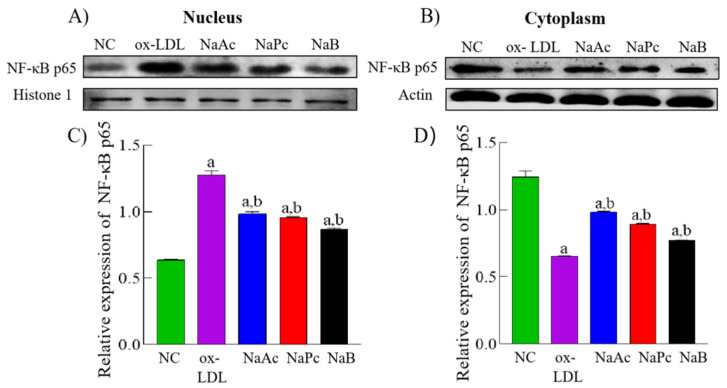
NF-κB p65 expressions in THP-1 cell nucleus and cytoplasm. The THP-1 cells were pre-treated with 50 mg/L ox-LDL for 24 h and then treated with 200 μmol/L NaAc, NaPc, and NaB for 24 h. The expressions of NF-κB p65 were determined by Western blotting assay. (**A**,**C**) are in the nucleus; (**B**,**D**) are in the cytoplasm. ^a^
*p* < 0.05 vs. NC group. ^b^
*p* < 0.05 vs. ox-LDL group.

**Figure 3 molecules-27-08801-f003:**
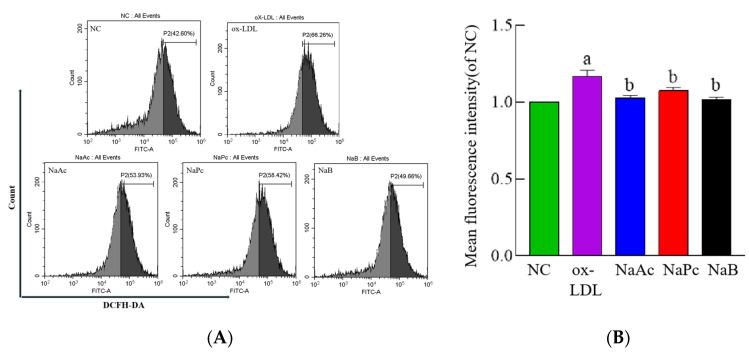
ROS production in THP-1 cells. The THP-1 cells were pre-treated with 50 mg/L ox-LDL for 24 h and then treated with 200 μmol/L NaAc, NaPc, and NaB for 24 h. ROS production was determined by flow cytometry. (**A**) The histogram of flow cytometry analysis. (**B**) The production of three main short-chain fatty acids. ^a^
*p* < 0.05 vs. NC group. ^b^
*p* < 0.05 vs. ox-LDL group.

**Figure 4 molecules-27-08801-f004:**
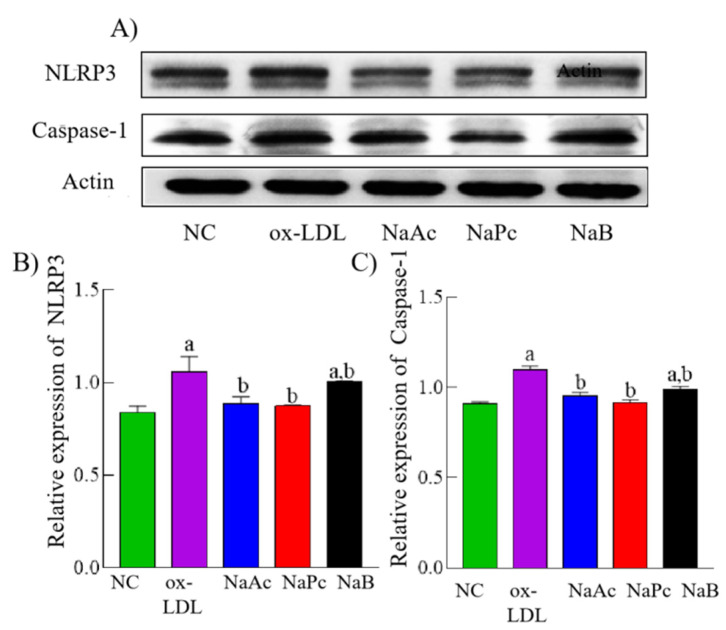
The expressions of NLRP3 and Caspase-1 in THP-1 cells. The THP-1 cells were pre-treated with 50 mg/L ox-LDL for 24 h and then treated with 200 μmol/L NaAc, NaPc, and NaB for 24 h. The expressions of NLRP3 and Caspase-1 were determined by Western blotting assay. (**A**) Representative images of protein expression. (**B**) The relative expression of NLRP3. (**C**) The relative expression of Caspase-1. ^a^
*p* < 0.05 vs. NC group. ^b^
*p* < 0.05 vs. ox-LDL group.

**Figure 5 molecules-27-08801-f005:**
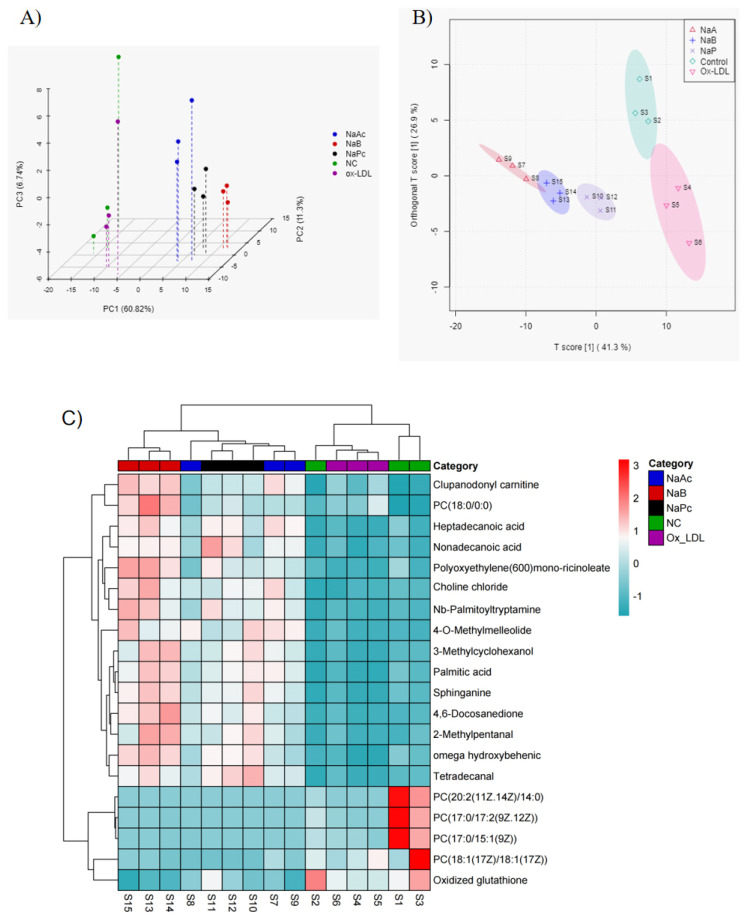
The PCA score plots, OPLS-DA score plots, and heat map of differential metabolites in THP-1 cells after being treated with the three main SCFAs. (**A**) PCA score plots; (**B**) OPLS-DA score plots; (**C**) heat map.

**Figure 6 molecules-27-08801-f006:**
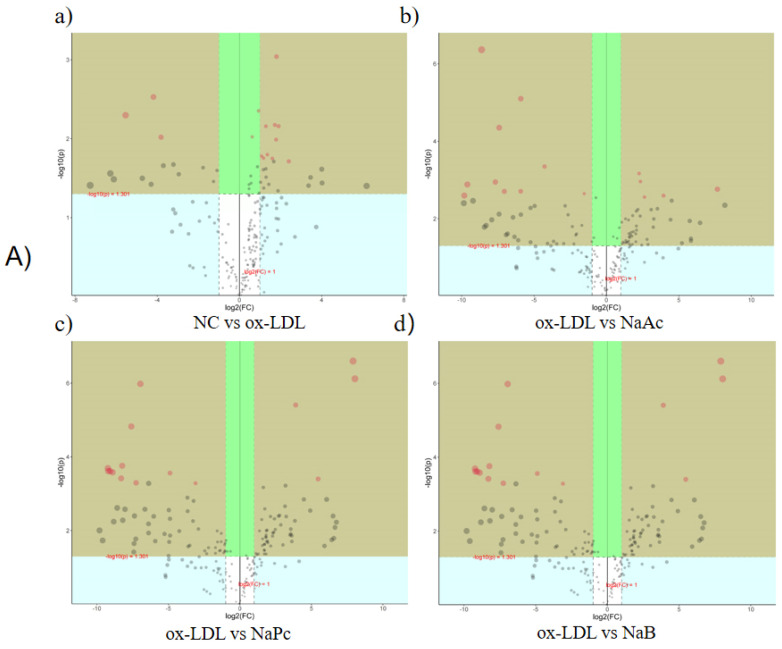
Volcanic diagram of pairwise comparison of differential metabolites and the metabolic pathways of THP-1 after treatment with the three main SCFAs.

**Figure 7 molecules-27-08801-f007:**
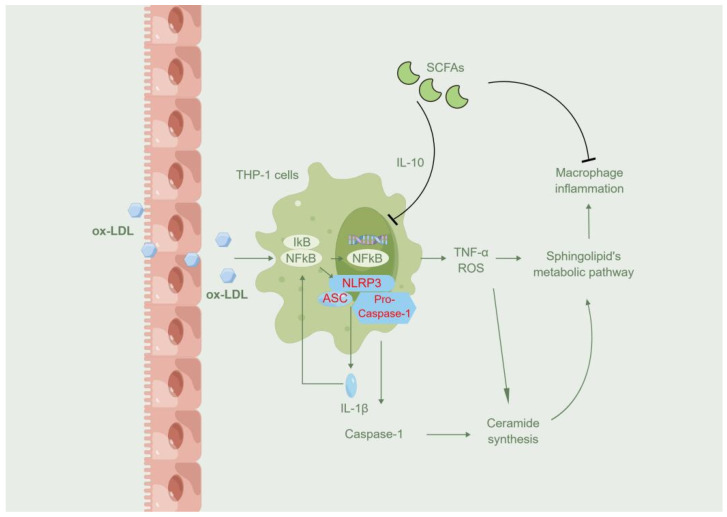
Schematic illustration of mechanism. Schematic illustration of the protective effects of SCFAs on ox-LDL-induced cell inflammatory injury via regulating the sphingolipid pathway and inhibiting the activation of the NLRP3/caspase-1 pathway.

**Table 1 molecules-27-08801-t001:** Potential differential metabolites between the ox-LDL and NaAc groups.

No.	*m/z*	Retention Time/min	Metabolite	VIP	Database Number	Formula	Trend
1	642.38	4.94	5-Methoxy-7-(4-hydroxyphenyl)-1-phenyl-3-heptanone	1.250	HMDB33294	C_20_H_24_O_3_	↓
2	724.53	23.35	PE(O-16:0/18:2(9Z,12Z))	1.249	LMGP02020100	C_39_H_76_NO_7_P	↓
3	750.54	23.35	PE(O-18:0/18:3(6Z,9Z,12Z))	1.250	LMGP02020049	C_41_H_78_NO_7_P	↓
4	857.27	4.26	Acemannan	1.234	HMDB40717	C_66_H_100_NO_49_-	↓
5	239.24	13.49	3-Methylcyclopentadecanone	1.221	HMDB34181	C_16_H_30_O	↑
6	418.39	5.42	Ethyl decanoate	1.217	HMDB30998	C_12_H_24_O_2_	↑
7	792.59	23.33	PC(O-16:0/20:3(8Z,11Z,14Z))	1.247	HMDB39527	C_44_H_84_NO_7_P	↓
8	390.30	8.32	PGE2alpha dimethyl amine	1.247	LMFA03010114	C_22_H_41_NO_3_	↑
9	280.26	9.26	2-(5,8-Tetradecadienyl)cyclobutanone	1.247	HMDB37519	C_18_H_30_O	↑
10	716.56	23.33	PC(P-14:0/18:1(9Z))	1.243	LMGP01030004	C_40_H_78_NO_7_P	↓
11	740.55	23.34	3-Oxohexadecanoic acid glycerides	1.244	HMDB39849	C_19_H_37_O_6−_	↓
12	819.24	4.26	Delphinidin 3-lathyroside 5-(6-acetylglucoside)	1.205	HMDB37091	C_34_H_41_O_22+_	↓
13	392.31	7.67	N-stearoyl GABA	1.244	LMFA08020106	C_22_H_43_NO_3_	↑
14	455.23	5.15	Glucosyl (2E,6E,10x)-10,11-dihydroxy-2,6-farnesadienoate	1.227	HMDB37823	C_21_H_36_O_9_	↑
15	256.26	11.60	Palmitic amide	1.243	HMDB12273	C_16_H_33_NO	↑

**Table 2 molecules-27-08801-t002:** Potential differential metabolites between the ox-LDL and NaPc groups.

No.	*m/z*	Retention Time /min	Metabolite	VIP	Database Number	Formula	Trend
1	642.38	4.94	5-Methoxy-7-(4-hydroxyphenyl)-1-phenyl-3-heptanone	1.225	HMDB33294	C_20_H_24_O_3_	↓
2	553.33	9.08	Lithocholate 3-O-glucuronide	1.225	HMDB02513	C_30_H_48_O_9_	↑
3	724.53	23.35	PE(O-16:0/18:2(9Z,12Z))	1.224	LMGP02020100	C_39_H_76_NO_7_P	↓
4	750.54	23.35	PE(O-18:0/18:3(6Z,9Z,12Z))	1.224	LMGP02020049	C_41_H_78_NO_7_P	↓
5	857.27	4.26	Acemannan	1.224	HMDB40717	C_66_H_100_NO_49−_	↓
6	857.57	4.26	PG(17:1(9Z)/22:0)	1.224	LMGP04010273	C_45_H_87_O_10_P	↓
7	857.67	4.26	Ergosterol peroxide	1.223	HMDB37941	C_28_H_44_O_3_	↓
8	244.26	4.94	Pentadecanal	1.222	HMDB31078	C_15_H_30_O	↑
9	390.30	8.32	PGE2alpha dimethyl amine	1.220	LMFA03010114	C_22_H_41_NO_3_	↑
10	792.59	23.33	1-O-Hexadecyl-2-O-dihomogammalinolenoylglycero-3-phosphocholine	1.219	HMDB39527	C_44_H_84_NO_7_P	↓
11	280.26	9.26	2-(5,8-Tetradecadienyl)cyclobutanone	1.219	HMDB37519	C_18_H_30_O	↑
12	740.55	23.34	3-Oxohexadecanoic acid glycerides	1.215	HMDB39849	C_19_H_37_O_6−_	↓
13	239.24	13.49	3-Methylcyclopentadecanone	1.215	HMDB34181	C_16_H_30_O	↑
14	716.56	23.33	PC(P-14:0/18:1(9Z))	1.213	LMGP01030004	C_40_H_78_NO_7_P	↓
15	308.09	4.74	Glutathione	1.213	HMDB00125	C_10_H_17_N_3_O_6_S	↓

**Table 3 molecules-27-08801-t003:** Potential differential metabolites between the ox-LDL and NaB groups.

No.	*m/z*	Retention Time /min	Metabolite	VIP	Database Number	Formula	Trend
1	686.10	1.04	Fenugreekine	1.167	HMDB39421	C_21_H_27_N_7_O_14_P_2_	↓
2	724.53	23.35	PE(O-16:0/18:2(9Z,12Z))	1.167	LMGP02020100	C_39_H_76_NO_7_P	↓
3	280.26	9.26	2-(5,8-Tetradecadienyl)cyclobutanone	1.167	HMDB37519	C_18_H_30_O	↑
4	724.53	23.35	PE(O-16:0/18:2(9Z,12Z))	1.166	LMGP02020100	C_39_H_76_NO_7_P	↓
5	642.38	4.94	5-Methoxy-7-(4-hydroxyphenyl)-1-phenyl-3-heptanone	1.166	HMDB33294	C_20_H_24_O_3_	↓
6	553.33	9.08	Lithocholate 3-O-glucuronide	1.166	HMDB02513	C_30_H_48_O_9_	↑
7	256.26	11.60	Palmitic amide	1.166	HMDB12273	C_16_H_33_NO	↑
8	819.24	4.26	Delphinidin 3-lathyroside 5-(6-acetylglucoside)	1.165	HMDB37091	C_34_H_41_O_22+_	↓
9	857.47	4.26	Quinquenoside F1	1.165	HMDB39398	C_42_H_74_O_15_	↓
10	857.37	4.26	(E)-Casimiroedine	1.165	HMDB30274	C_21_H_27_N_3_O_6_	↓
11	857.27	4.26	Acemannan	1.165	HMDB40717	C_66_H_100_NO_49−_	↓
12	857.57	4.26	PG(17:1(9Z)/22:0)	1.165	LMGP04010273	C_45_H_87_O_10_P	↓
13	655.33	5.67	Argenteane	1.164	HMDB39454	C_40_H_46_O_8_	↑
14	455.23	5.15	Glucosyl (2E,6E,10x)-10,11-dihydroxy-2,6-farnesadienoate	1.164	HMDB37823	C_21_H_36_O_9_	↑
15	857.67	4.26	Ergosterol peroxide	1.164	HMDB37941	C_28_H_44_O_3_	↓

## Data Availability

The data presented in this study are available upon request from the corresponding authors.

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
