# Peer review of "Short-Chain Fatty Acids Weaken Ox-LDL-Induced Cell Inflammatory Injury by Inhibiting the NLRP3/Caspase-1 Pathway and Affecting Cellular Metabolism in THP-1 Cells"

_molecules, 2022, doi:10.3390/molecules27248801_

Round 1
Reviewer 1 Report
The manuscript by Yi et al reported on the analysis of short-chain fatty acids (SCFAs) on the ox-LDL dependent inflammatory pathways related to atherosclerosis. While the work was outside my field of expertise, overall, the work seemed sound. Some minor adjustments and perhaps a few control experiments would be helpful prior to publication.
1) At times, it was a bit difficult to follow the work. For the broad readership of molecules, a clearer outline of the introduction and discussion would be welcomed. This could be achieved through the addition of an introductory/cartoon figure.
2) Some of the integrated gel images have very small uncertainties, which seem unrealistic (e.g., Figure 2). It would be useful to show other representatives in an SI to support such results.
3) While the authors justify 200 microM concentrations of these "SCFAs", it would be helpful to see a dose-dependence effect. What is the threshold? Perhaps on two different assays, such as the initial qPCR and Nf-kB expression, where the impacts are reported to be most significant.
4) Have the authors compared a normal FA for example, palmitic acid, to see if similar (or more significant) effects are seen just to clarify if the behavior reported in the paper can be solely attributed to short-chain versions of FAs?
Author Response
Reviewer 1#
The manuscript by Yi et al reported on the analysis of short-chain fatty acids (SCFAs) on the ox-LDL dependent inflammatory pathways related to atherosclerosis. While the work was outside my field of expertise, overall, the work seemed sound. Some minor adjustments and perhaps a few control experiments would be helpful prior to publication.
- At times, it was a bit difficult to follow the work. For the broad readership of molecules, a clearer outline of the introduction and discussion would be welcomed. This could be achieved through the addition of an introductory/cartoon figure.
Response: Thanks for your comments. We added a introductory figure to show the schematic illustration of mechanism.
- Some of the integrated gel images have very small uncertainties, which seem unrealistic (e.g., Figure 2). It would be useful to show other representatives in an SI to support such results.
Response: All the Western blotting assays were repeated at least 3 times, and the differences between the groups were compared by one-way ANOVA. The analysis of the expression of p65 protein in the nucleus and cytoplasm is a common technique used by our research group to study the activation of NF-κB pathway, and this technique is also a common method. We plan to further study the phosphorylation of related proteins in the NF-κB signaling pathway in order to obtain more convincing results.
3) While the authors justify 200 microM concentrations of these "SCFAs", it would be helpful to see a dose-dependence effect. What is the threshold? Perhaps on two different assays, such as the initial qPCR and Nf-kB expression, where the impacts are reported to be most significant.
Response: The concentration of 200 mM is the appropriate concentration we have explored in the previous study. The concentration of THP-1 cells and other cells did not have obvious toxicity
- Have the authors compared a normal FA for example, palmitic acid, to see if similar (or more significant) effects are seen just to clarify if the behavior reported in the paper can be solely attributed to short-chain versions of FAs?
Response: Your suggestion is very good. This is something that we didn't take into account in the design of the experiment. In this paper, we want to study the function of short-chain fatty acids, a special metabolite of intestinal flora, but we plan to add palmitic acid in the next stage of research.
Reviewer 2 Report
This paper is of important significance due to anti-inflammatory and anti-oxidative effects of SCFAs produced by intestinal microbiota. The importance of this reserch is high because the association between SCFAs and ox-LDL-induced macrophage activation with respect to the formation of atherosclerosis plaque has not yet been elucidated. It was demonstrated that SCFAs could significantly inhibit cell inflammatory injury induced by ox-LDL. Moreover, SCFAs decreased the expression of IL-1β and TNF-α via inactivation of the NLRP3/Caspase-1 pathway induced by ox‑LDL. Furthermore, SCFAs affected cellular metabolism in ox‑LDL‑induced macrophages, as detected by untargeted metabolomics analysis. These findings may provide novel insights into the role of SCFAs intervention in the progression of atherosclerotic plaque formation. The paper sounds scientific, brings new findings and is worthg of publication.
Author Response
Reviewer 2#
This paper is of important significance due to anti-inflammatory and anti-oxidative effects of SCFAs produced by intestinal microbiota. The importance of this reserch is high because the association between SCFAs and ox-LDL-induced macrophage activation with respect to the formation of atherosclerosis plaque has not yet been elucidated. It was demonstrated that SCFAs could significantly inhibit cell inflammatory injury induced by ox-LDL. Moreover, SCFAs decreased the expression of IL-1β and TNF-α via inactivation of the NLRP3/Caspase-1 pathway induced by ox‑LDL. Furthermore, SCFAs affected cellular metabolism in ox‑LDL‑induced macrophages, as detected by untargeted metabolomics analysis. These findings may provide novel insights into the role of SCFAs intervention in the progression of atherosclerotic plaque formation. The paper sounds scientific, brings new findings and is worthg of publication.
Response: Thanks for your comments.
Reviewer 3 Report
The immunomodulatory potential of gut microbiome-derived short-chain fatty acids (SCFAs) has gained significant interest. In this study, the authors found that SCFAs (sodium acetate, sodium propionate, and sodium butyrate) can weaken ox-LDL-induced cell inflammatory injury by inhibiting NLRP3/caspase-1 pathway and affecting cellular metabolism in THP-1 cells. I think the results will contribute to improve understanding of the role of SCFAs intervention in chronic inflammatory disease. It is recommended for possible publication in Molecules after the following issues are addressed.
1. How to select the concentration of three SCFAs (NaAc, NaPc and NaB)? Another article by the authors (doi: 10.3639/j. issn. 1009-5551.2022.01.006) studied that sodium butyrate at different concentrations inhibited ox-LDL induced THP-1 cell inflammatory response. Please discuss the influences of treatment time and concentration of SCFAs on the inhibition of ox-LDL induced THP-1 cell inflammation.
2. The author said that the anti-inflammatory mechanisms were studied by untargeted metabolomics techniques. However the association between the cellular metabolites and the anti-inflammatory mechanism is not convincing.
In Figure 5A the grouping in PCA shows that there is no significant separation between the NC and ox-LDL groups on PC1.
Line 162,FC > 1 as the boundary?
Line 215 reduced glutathione (GSH) was significantly up-regulated in THP-1 cells, the GSH data is absent.
Line 255-256 sphingolipids metabolic pathway of THP-1 cells was activated by ox-LDL in vitro and inhibited by the three SCFAs. In Figure 6B, sphingolipid metabolism is significant only in ox LDL vs NaPc, which -logP reaches the threshold of 1.3.
The table of differential metabolites between the NC vs. ox-LDL groups is missing.
Section 4.5. Cellular metabolites analysis, UPLC-QTof-MS/MS conditions is missing.
3. It is interesting to note that butyrate is significantly different from acetate and propionate, in the expressions of pro-inflammation cytokines, regulation of the expression of NLRP3/Caspase-1 pathway, as well as the metabolite levels (Figure 5C heat map). Further data mining is recommended.
Author Response
Reviewer 3#
The immunomodulatory potential of gut microbiome-derived short-chain fatty acids (SCFAs) has gained significant interest. In this study, the authors found that SCFAs (sodium acetate, sodium propionate, and sodium butyrate) can weaken ox-LDL-induced cell inflammatory injury by inhibiting NLRP3/caspase-1 pathway and affecting cellular metabolism in THP-1 cells. I think the results will contribute to improve understanding of the role of SCFAs intervention in chronic inflammatory disease. It is recommended for possible publication in Molecules after the following issues are addressed.
- How to select the concentration of three SCFAs (NaAc, NaPc and NaB)? Another article by the authors (doi: 10.3639/j. issn. 1009-5551.2022.01.006) studied that sodium butyrate at different concentrations inhibited ox-LDL induced THP-1 cell inflammatory response. Please discuss the influences of treatment time and concentration of SCFAs on the inhibition of ox-LDL induced THP-1 cell inflammation.
Response: This concentration was used by our research group for 2 years. After repeated trials in previous studies, we found that 100-200 μmol/L concentration had the best inhibitory effect on inflammation, and no obvious cytotoxicity. Since THP-1 cells are a type of macrophage, they cannot be maintained for more than 48 h after macrophage.
- The author said that the anti-inflammatory mechanisms were studied by untargeted metabolomics techniques. However the association between the cellular metabolites and the anti-inflammatory mechanism is not convincing.
Response: Thanks for your comments. Metabonomics provides new possibilities for our research from the perspective of cell metabolism. We have added relevant discussions and shortcomings of this study.
In Figure 5A the grouping in PCA shows that there is no significant separation between the NC and ox-LDL groups on PC1.
Response: The OPLS-DA score plots showed that the NC group was completely separated from the ox-LDL group.
Line 162,FC > 1 as the boundary?
Response: I'm terribly sorry. It was a mistake. Should be corrected to VIP. Revised.
Line 215 reduced glutathione (GSH) was significantly up-regulated in THP-1 cells, the GSH data is absent.
Response: We made a mistake when we tried to count them, and we have corrected them in the revised manuscript. Thanks for your comments.
Line 255-256 sphingolipids metabolic pathway of THP-1 cells was activated by ox-LDL in vitro and inhibited by the three SCFAs. In Figure 6B, sphingolipid metabolism is significant only in ox LDL vs NaPc, which -logP reaches the threshold of 1.3.
Response: The revised manuscript has been corrected because we made a mistake counting them. Thanks for your comments.
The table of differential metabolites between the NC vs. ox-LDL groups is missing.
Response: This table has been omitted since this paper is primarily interested in three short-chain fatty acids.
Section 4.5. Cellular metabolites analysis, UPLC-QTof-MS/MS conditions is missing.
Response: The relevant content has been added to the revised manuscript.
- It is interesting to note that butyrate is significantly different from acetate and propionate, in the expressions of pro-inflammation cytokines, regulation of the expression of NLRP3/Caspase-1 pathway, as well as the metabolite levels (Figure 5C heat map). Further data mining is recommended.
Response: Your opinions are very valuable to our study, and we will dig into the data in the next stage of research.
Reviewer 4 Report
1. Format of the figures are inconsistent throughout the manuscript. Figures that are composed of more than one sub-figures should be numbered as A, B, C, etc., and the numbering style should be consistent (for example: sub-figures in Figure 3 lacks numbering).
2. This manuscript needs extensive revision for language and grammar before publication.
Author Response
Reviewer 4#
- Format of the figures are inconsistent throughout the manuscript. Figures that are composed of more than one sub-figures should be numbered as A, B, C, etc., and the numbering style should be consistent (for example: sub-figures in Figure 3 lacks numbering).
Response: Thanks for your comments. We had modified.
- This manuscript needs extensive revision for language and grammar before publication.
Response: The revised manuscript has been copyedited by a language editor who is a native English speaker with extensive scientific/technical background.
Round 2
Reviewer 1 Report
The authors have taken efforts to improve readability and incorporated a new summary figure (Fig 7) that does help to support the larger picture. This reviewer still has concerns about the level of error/uncertainty reported. More details in the methods should be expanded and perhaps show the three sets of data in the SI to support the error.